# DNA Code from Cyclic and Skew Cyclic Codes over $F_{4}\left[v\right]/\langle v^{3}\rangle$

**DOI:** 10.3390/e25020239

**Published:** 2023-01-28

**Authors:** Om Prakash, Ashutosh Singh, Ram Krishna Verma, Patrick Solé, Wei Cheng

**Affiliations:** 1Department of Mathematics, Indian Institute of Technology Patna, Bihar 801106, India; 2Department of Mathematics, SRM Institute of Science and Technology, Delhi-NCR Campus, Ghaziabad 201204, India; 3I2M, (CNRS, Aix-Marseille University, Centrale Marseille), 13009 Marseilles, France; 4LTCI, Télécom Paris (IP Paris), 19 Place Marguerite Perey, 91120 Palaiseau, France; 5Secure-IC S.A.S., 104 Boulevard du Montparnasse, 75014 Paris, France

**Keywords:** reversible code, gray map, DNA codes

## Abstract

The main motivation of this work is to study and obtain some reversible and DNA codes of length *n* with better parameters. Here, we first investigate the structure of cyclic and skew cyclic codes over the chain ring $R:=F_{4}\left[v\right]/\langle v^{3}\rangle$. We show an association between the codons and the elements of R using a Gray map. Under this Gray map, we study reversible and DNA codes of length *n*. Finally, several new DNA codes are obtained that have improved parameters than previously known codes. We also determine the Hamming and the Edit distances of these codes.

## 1. Introduction

DNA is a nucleic acid used for carrying genetic information in living organisms. It is a double-strand molecule formed from two possible nitrogenous bases—Purines (Adenine and Guanine) and Pyrimidines (Cytosine—and Thymine) and two chemically polar ends, namely, $5^{\prime }$ and $3^{\prime }$. The Watson–Crick complementary (WCC) relation, which is characterized as $A^{c}=T,\;G^{c}=C$, and vice versa, is used to bind the bases of DNA. In 1994, Adleman [1] discussed the Hamiltonian path problem using DNA molecules. This (NP-complete) problem is solved by encoding a small graph in DNA molecules where all the operations were carried out using standard protocols such as the WCC relation. Due to massive parallelism, DNA computing emerged as a powerful tool among researchers to solve computationally difficult problems. Further, the experiments are performed on synthesized DNA and RNA molecules to control their combinatorial constraints such as constant GC-content and Hamming distance.

Linear codes over finite fields have been explored for almost three decades, but this research area experienced an astonishing rate after the remarkable work of Hammons et al. [2] when they established a relation between linear codes over $Z_{4}$ with other non-linear binary codes. Afterward, many authors [3,4,5,6] considered alphabets endowed with a ring structure and found many good linear codes over finite fields via specific Gray maps. Within the class of linear codes, cyclic codes are the pivotal and the most studied codes due to their theoretical richness and practical implementation. Recently, many authors [7,8,9,10,11,12,13] constructed DNA codes using cyclic codes over rings. For instance, Bayram et al. [7] and Yildiz and Siap [13] explored DNA codes over the rings $F_{4}+vF_{4}$, $v^{2}=v$ and $F_{2}\left[v\right]/\langle v^{4}-1\rangle$, respectively. In 2019, Mostafanasab and Darani [12] discussed the structure of cyclic DNA codes over the chain ring $F_{2}+uF_{2}+u^{2}F_{2}$. Liu et al. [14] worked on cyclic DNA codes of an odd length over $F_{4}\left[u\right]/\langle u^{3}\rangle$. On the other hand, Boucher et al. [15] introduced skew cyclic codes and discovered many new linear codes. Further, in [16,17], more properties of these codes over chain rings have been established. Recently, Gursoy et al. [18] studied reversible DNA codes by using skew cyclic codes. Later on, Cengellenmis et al. [19] studied DNA codes from skew cyclic codes over the rings $F_{2}\left[u,v,w\right],\;\mathrm{where}\;u^{2}=v^{2}+v=w^{2}+w=uv+vu=uw+wu=vw+wv=0$. Motivated by the above works, we consider cyclic as well as skew cyclic codes over the finite chain ring $R=F_{4}\left[v\right]/\langle v^{3}\rangle$ to construct DNA codes of arbitrary lengths. Hamming and edit distances are also calculated for the obtained codes. Interestingly, we obtain several new codes with better parameters than known codes [14].

The article is structured as follows: The Gray map, together with the correspondence of the codons and the other basic results of cyclic codes, are in Section 2. Reversible cyclic codes over the ring R are covered in Section 3, whereas the reversible skew cyclic codes are studied in Section 4. Some results related to the complement and reverse complement of obtained codes are presented in Section 5. Based on our established results from the previous Sections and magma computer algebra system [20], we provide a few examples of DNA codes of arbitrary lengths in Section 6. In the end, we conclude our work in Section 7.

## 2. Preliminaries

Let $F_{4}=\left\{0,1,t,t^{2}\right\},$ where $t^{2}=t+1$ be a finite field. Then $R:=F_{4}\left[v\right]/\langle v^{3}\rangle$ is a finite chain ring with characteristic 2 and every element *r* of R can be represented as $r=b_{1}+b_{2}v+b_{3}v^{2}$ where $b_{i}\in F_{4},$ for i=0,1,2 and $v^{3}=0$. It is easy to show that R is a principal ideal ring with unique maximal ideal 〈vs.〉 and R/〈vs.〉 is isomorphic to $F_{4}$. Recall that the ring R has 48 invertible elements of the form $r=b_{1}+b_{2}v+b_{3}v^{2}$, where $b_{1}$ is invertible in $F_{4}$.

A linear code C of length *n* and alphabets from R is a submodule of an R-module $R^{n}$. The elements of C are called the codewords. The Hamming weight of an element $b=\left(b_{0},b_{1},\cdots ,b_{n}\right)\in C$ is defined as $w_{H}\left(b\right)$= $\left|\{i|b_{i}\neq 0\}\right|$ and Hamming distance $d_{H}\left(b,k\right)$ between any two elements $b=(b_{0},b_{1},\cdots ,b_{n})$ and $k=(k_{0},k_{1},\cdots ,k_{n})$ in C is defined as $d_{H}\left(b,k\right)=w_{H}\left(b-k\right).$ Additionally, the lowest value in the set $\left\{d_{H}\left(b,k\right)|b\neq k,\;\forall \;b,k\in C\right\}$ is considered as the the Hamming distance $d_{H}\left(C\right)$ of the code C.

Now, we describe a Gray map $\Phi :R⟶F_{4}^{3}$ as:(1)$$\Phi \left(b_{0}+b_{1}v+b_{2}v^{2}\right)=\left(b_{0}+b_{1}+b_{2},b_{1}+b_{2},b_{2}\right),$$
where $b_{i}\in F_{4}$ for i=0,1,2. It is easy to see that the function Φ is a distance-preserving map and is extendable to $R^{n}$ component-wise. In Table 1, we establish the connection between the ring elements and the codons by using the Gray map (Equation 1).

**Definition** **1.**
*For a given polynomial $g\left(z\right)=g_{0}+g_{1}z+\ldots +g_{m}z^{m}\in F_{4}\left[z\right]$, the reciprocal polynomial is denoted by $g^{*}\left(z\right)$ and defined as $g^{*}\left(z\right)=\sum_{i=0}^{m}g_{m-i}z^{i}$. A polynomial g(z) is said to be self-reciprocal if and only if $g^{*}\left(z\right)=bg\left(z\right)$ for some non-zero element b in $F_{4}$.*


Now, we present some useful lemmas that appeared in [8,14].

**Lemma** **1.**
*Let g(z) and h(z) be polynomials over R of degrees r and s, respectively, with r≥s. Then:*
*1*. 

${\left[g\left(z\right)h\left(z\right)\right]}^{*}=g^{*}\left(z\right)h^{*}\left(z\right)$

*2*. 
*${\left[g\left(z\right)+h\left(z\right)\right]}^{*}=g^{*}\left(z\right)+z^{\left(r-s\right)}h^{*}\left(z\right)$.*



**Lemma** **2.**
*Let f(z), g(z), and h(z) be polynomials over R of degrees r, s, and t, respectively, where r≥s,t. Then:*
*1*. 

${\left[f\left(z\right)g\left(z\right)h\left(z\right)\right]}^{*}=f^{*}\left(z\right)g^{*}\left(z\right)h^{*}\left(z\right)$

*2*. 
*${\left[f\left(z\right)+g\left(z\right)+h\left(z\right)\right]}^{*}=f^{*}\left(z\right)+z^{\left(r-s\right)}g^{*}\left(z\right)+z^{\left(r-t\right)}h^{*}\left(z\right)$.*



Using the Watson–Crick complementary relation, we define the reverse (**R**) and the reverse complement (**RC**) of a DNA codeword $b=(b_{0},b_{1},\ldots ,b_{n-1})$ by $b^{r}=\left(b_{n-1},\ldots ,b_{1},b_{0}\right)$ and $b^{rc}=\left(b_{n-1}^{c},\ldots ,b_{1}^{c},b_{0}^{c}\right)$, respectively. For example, given b=ATCCGT, we obtain $b^{r}=TGCCTA$ and $b^{rc}=ACGGAT$.

We have the following observations based on the Gray map provided in Equation (Equation 1).

**Lemma** **3.**
*1*. 
*For any $a=(b_{0}+b_{1}v+b_{2}v^{2})\in R$, we have*

*$\Phi {\left(b_{0}+b_{1}v+b_{2}v^{2}\right)}^{r}=b_{1}+b_{0}v+\left(b_{0}+b_{1}+b_{2}\right)v^{2},$ where $b_{0},b_{1},b_{2}\in F_{4}.$*
*2*. 
*$\Phi {\left(b_{0}+b_{1}\right)}^{r}=\Phi {\left(b_{0}\right)}^{r}+\Phi {\left(b_{1}\right)}^{r},$ where $b_{0},b_{1}\in F_{4}$.*



## 3. Reversible Cyclic Codes over R

In the present section, we investigate the structure of cyclic codes and prove reversible conditions on these codes. The cyclic codes of odd lengths are provided in [14] and a detailed discussion on cyclic codes of arbitrary length with alphabets from $Z_{2}\left[u\right]/\langle v^{3}\rangle$ is explored in [6]. Now, in the subsequent theorems, we describe the structure of the cyclic code. We omit the proof due to its similarity to the proof provided in [6].

**Theorem** **1.**
*Let C be a cyclic code of length n over R. Then the code C is provided by:*

$$C=\langle g_{0}\left(z\right)+vg_{1}\left(z\right)+v^{2}g_{2}\left(z\right),va_{1}\left(z\right)+v^{2}p\left(z\right),v^{2}a_{2}\left(z\right)\rangle$$

*where $a_{2}\left(z\right)|a_{1}\left(z\right)\left|g_{0}\left(z\right)\right|\left(z^{n}-1\right)$ over $F_{4}$, $a_{1}\left(z\right)|g_{1}\left(z\right)\left(\frac{z^{n}-1}{g_{0}\left(z\right)}\right)$, $a_{2}\left(z\right)|p\left(z\right)\left(\frac{z^{n}-1}{a_{1}\left(z\right)}\right)$, and $a_{2}\left(z\right)|g_{2}\left(z\right)$$\left(\frac{z^{n}-1}{g_{0}\left(z\right)}\right)\left(\frac{z^{n}-1}{a_{1}\left(z\right)}\right)$ over $F_{4}$. Moreover, $deg\left(g_{2}\left(z\right)\right)<deg\left(a_{2}\left(z\right)\right)$, $deg\left(p\left(z\right)\right)<deg\left(a_{2}\left(z\right)\right)$, and $deg\left(g_{1}\left(z\right)\right)<deg\left(a_{1}\left(z\right)\right)$.*


**Corollary** **1.**
*If the length of a cyclic code C is odd and $g_{1}\left(z\right)=g_{2}\left(z\right)=p\left(z\right)=0$, then $C=\langle g_{0}\left(z\right),va_{1}\left(z\right),v^{2}a_{2}\left(z\right)\rangle =\langle g_{0}\left(z\right)+va_{1}\left(z\right)+v^{2}a_{2}\left(z\right)\rangle$.*


A similar result is also possible when *n* is not odd. In this case, we assume that $\gcd(\frac{z^{n}-1}{a_{2}\left(z\right)},g_{0}\left(z\right))=1$ and consequently obtain the following result.

**Corollary** **2.**
*If a cyclic code C is of even length n and $\gcd(\frac{z^{n}-1}{a_{2}\left(z\right)},g_{0}\left(z\right))=1$, then $g_{1}\left(z\right)=g_{2}\left(z\right)=p\left(z\right)=0$.*


When $a_{2}\left(z\right)=g_{0}\left(z\right)$, then $a_{2}\left(z\right)=a_{1}\left(z\right)=g_{0}\left(z\right)$ and C as a subset of $\langle g_{0}\left(z\right)+vg_{1}\left(z\right)+v^{2}g_{2}\left(z\right)\rangle$. Since the other containment is true by the definition of C, we, therefore, obtain the following corollary.

**Corollary** **3.**
*For a cyclic code $C=\langle g_{0}\left(z\right)+vg_{1}\left(z\right)+v^{2}g_{2}\left(z\right),va_{1}+v^{2}p\left(z\right),v^{2}a_{2}\left(z\right)\rangle$, if $a_{2}\left(z\right)=g_{0}\left(z\right)$, then $C=\langle g_{0}\left(z\right)+vg_{1}\left(z\right)+v^{2}g_{2}\left(z\right)\rangle$.*


**Definition** **2.**
*Given a code $C=\langle g_{0}\left(z\right)+vg_{1}\left(z\right)+v^{2}g_{2}\left(z\right),va_{1}\left(z\right)+v^{2}p\left(z\right),v^{2}a_{2}\left(z\right)\rangle$ over R, we define $C_{v^{2}}$ by $\left\{q\left(z\right)\in F_{4}\left[z\right]\;|\;v^{2}q\left(z\right)\in C\right\}$. Particularly, since $a_{2}\left(z\right)\left|a_{1}\left(z\right)\right|g_{0}\left(z\right)$, $C_{v^{2}}=\langle a_{2}\left(z\right)\rangle$.*


In the next result, we determine the Hamming distance of the code C by using the above definition in terms of the Hamming distance of $C_{v^{2}}$.

**Theorem** **2.**
*Let C be a code provided by $C=\langle g\left(z\right)+vg_{1}\left(z\right)+v^{2}g_{2}\left(z\right),va_{1}\left(z\right)+v^{2}p\left(z\right),v^{2}a_{2}\left(z\right)\rangle$. Then Hamming distance of C and $C_{v^{2}}$ are equal, i.e., $d_{H}\left(C\right)=d_{H}\left(C_{v^{2}}\right).$*


**Proof.** It can be obtained from [4]. □

**Remark** **1.**
*For the sake of brevity, we use b for polynomial b(z) whenever b(z) belongs to the field $F_{4}$.*


**Lemma** **4.**
*Let $g_{0}\left(z\right),g_{1}\left(z\right)\;\mathrm{and}\;g_{2}\left(z\right)\in F_{4}\left[z\right]$ of degrees r,sandt, respectively. Then ${\left(g_{0}\left(z\right)+vg_{1}\left(z\right)+v^{2}g_{2}\left(z\right)\right)}^{*}=g_{0}^{*}\left(z\right)+vz^{r-s}g_{2}^{*}\left(z\right)+v^{2}z^{r-t}g_{2}^{*}\left(z\right)$.*


**Theorem** **3.**
*Let $C=\langle g_{0}\left(z\right)+vg_{1}\left(z\right)+v^{2}g_{2}\left(z\right)\rangle$ be a cyclic code of even length over R with monic polynomials $g_{0}\left(z\right)$, $g_{1}\left(z\right)$ and $g_{2}\left(z\right)$ of degrees r,s and t, respectively. Then the code C is reversible if and only if:*
*(1)* 
*$g_{0}\left(z\right)$ is a self-reciprocal polynomial;*
*(2)* 
*$z^{r-s}g_{1}^{*}\left(z\right)=b_{0}g_{1}\left(z\right)+b_{1}g_{0}\left(z\right)$ and $z^{r-s}g_{2}^{*}\left(z\right)=b_{0}g_{2}\left(z\right)+b_{1}g_{1}\left(z\right)+b_{2}g_{0}\left(z\right)$, where $b_{0}\in F_{4}\backslash \left\{0\right\}$ and $b_{1},b_{2}\in F_{4}$.*



**Proof.** Let C be a reversible cyclic code. Then
$$\begin{array}{cc} {\left(g_{0}\left(z\right)+vg_{1}\left(z\right)+v^{2}g_{2}\left(z\right)\right)}^{*} & =g_{0}^{*}\left(z\right)+vz^{r-s}g_{2}^{*}\left(z\right)+v^{2}z^{r-t}g_{2}^{*}\left(z\right)\;\mathrm{and}\\ {\left(g_{0}\left(z\right)+vg_{1}\left(z\right)+v^{2}g_{2}\left(z\right)\right)}^{*} & =b\left(z\right)(g_{0}\left(z\right)+vg_{1}\left(z\right)+v^{2}g_{2}\left(z\right))\in C\\ \; & =\left(b_{0}\left(z\right)+vb_{1}\left(z\right)+v^{2}b_{2}\left(z\right)\right)\left(g_{0}\left(z\right)+vg_{1}\left(z\right)+v^{2}g_{2}\left(z\right)\right)\\ \; & =b_{0}\left(z\right)g_{0}\left(z\right)+v\left(b_{0}\left(z\right)g_{1}\left(z\right)+b_{1}\left(z\right)g_{0}\left(z\right)\right)\\ \; & +v^{2}\left(b_{0}\left(z\right)g_{2}\left(z\right)+b_{1}\left(z\right)g_{1}\left(z\right)+b_{2}\left(z\right)g_{0}\left(z\right)\right). \end{array}$$Comparing right side of the two equations, we obtain $g_{0}^{*}\left(z\right)=b_{0}\left(z\right)g_{0}\left(z\right)$, $z^{r-s}g_{1}^{*}\left(z\right)=b_{0}\left(z\right)g_{1}\left(z\right)+b_{1}\left(z\right)g_{0}\left(z\right)$ and $z^{r-t}g_{2}^{*}\left(z\right)=b_{0}\left(z\right)g_{2}\left(z\right)+b_{1}\left(z\right)g_{1}\left(z\right)+b_{2}\left(z\right)g_{0}\left(z\right)$. Now, using $\deg f^{*}\left(z\right)\leq \deg f\left(z\right)$, we obtain $b_{0}\left(z\right)\neq 0$ in $F_{4}$ and this implies that the polynomial $g_{0}\left(z\right)$ is self-reciprocal. Therefore, $z^{r-s}g_{1}^{*}\left(z\right)=b_{0}g_{1}\left(z\right)+b_{1}\left(z\right)g_{0}\left(z\right)$ where $b_{0}=b_{0}\left(z\right)$ is a non-zero element in $F_{4}$. Now comparing the degrees of both sides, we obtain a constant polynomial $b_{1}\left(z\right)\in F_{4}$, say, $b_{1}$. We have $z^{r-t}g_{2}^{*}\left(z\right)=b_{0}g_{2}\left(z\right)+b_{1}g_{1}\left(z\right)+b_{2}\left(z\right)g_{0}\left(z\right)$. Again, comparing the degrees of both sides, we obtain $b_{2}\left(z\right)$ in $F_{4}$, say $b_{2}$. Thus, $z^{r-s}g_{1}^{*}\left(z\right)=b_{0}g_{1}\left(z\right)+b_{1}g_{0}\left(z\right)$ and $z^{r-t}g_{2}^{*}\left(z\right)=b_{0}g_{2}+b_{1}g_{1}\left(z\right)+b_{2}g_{0}\left(z\right)$ where $b_{0}\in F_{4}\backslash \left\{0\right\}$ and $b_{1},b_{2}\in F_{4}$.Conversely, assume (1) and (2) hold. Then
$$\begin{array}{cc} {\left(g_{0}\left(z\right)+vg_{1}\left(z\right)+v^{2}g_{2}\left(z\right)\right)}^{*}= & g_{0}^{*}\left(z\right)+vz^{r-s}g_{1}^{*}\left(z\right)+v^{2}z^{r-t}g_{2}^{*}\left(z\right)\\ = & b_{0}g_{0}\left(z\right)+vb_{0}g_{1}\left(z\right)+vb_{1}g_{0}\left(z\right)+v^{2}b_{0}g_{2}\left(z\right)\\ & +v^{2}b_{1}g_{1}\left(z\right)+v^{2}b_{2}g_{0}\left(z\right)\\ = & b_{0}\left(g_{0}\left(z\right)+vg_{1}\left(z\right)+v^{2}g_{2}\left(z\right)\right)+b_{1}\left(vg_{0}+v^{2}g_{1}\right)\\ & +b_{2}\left(v^{2}g_{0}\left(z\right)\right)\in C \end{array}$$Thus, the code C is reversible. □

**Theorem** **4.**
*Let $C=\langle g_{0}\left(z\right)+vg_{1}\left(z\right)+v^{2}g_{2}\left(z\right),v^{2}a_{2}\left(z\right)\rangle$ be a cyclic code of even length n over R with polynomials $g_{0}\left(z\right)$, $g_{1}\left(z\right)$, and $g_{2}\left(z\right)$ of degrees r,s, and t, respectively, and r>max{s,t}. Furthermore, assume that $a_{2}\left(z\right)\left|g_{0}\left(z\right)\right|\left(z^{n}-1\right)$. Then the code C is reversible if and only if:*
*(1)* 
*$g_{0}\left(z\right)$ and $a_{2}\left(z\right)$ are self-reversible;*
*(2)* 
*$z^{r-s}g_{1}^{*}\left(z\right)=b_{0}g_{1}\left(z\right)+b_{1}g_{0}\left(z\right)$, and $a_{2}\left(z\right)|(z^{r-t}g_{2}^{*}\left(z\right)+b_{0}g_{2}\left(z\right)+b_{1}g_{1}\left(z\right),$ where $b_{0}\in F_{4}\backslash \left\{0\right\}$ and $b_{1}\in F_{4}$.*



**Proof.** Let C be a reversible code. Then
$$\begin{array}{cc} {\left(g_{0}\left(z\right)+vg_{1}\left(z\right)+v^{2}g_{2}\left(z\right)\right)}^{*} & =g_{0}^{*}\left(z\right)+vz^{r-s}g_{1}^{*}\left(z\right)+v^{2}z^{r-t}g_{2}^{*}\left(z\right). \end{array}$$Furthermore,
$$\begin{array}{cc} {\left(g_{0}\left(z\right)+vg_{1}\left(z\right)+v^{2}g_{2}\left(z\right)\right)}^{*} & =b\left(z\right)\left(g_{0}\left(z\right)+vg_{1}\left(z\right)+v^{2}g_{2}\left(z\right)\right)+v^{2}c\left(z\right)a_{2}\left(z\right)\\ \; & =\left(b_{0}\left(z\right)+vb_{1}\left(z\right)+v^{2}b_{2}\left(z\right)\right)(g_{0}\left(z\right)+vg_{1}\left(z\right)+\\ \; & v^{2}g_{2}\left(z\right))+v^{2}c\left(z\right)a_{2}\left(z\right)\;\mathrm{where}\;b_{i}\left(z\right),\;c\left(z\right)\in F_{4}\left[z\right]\\ \; & =b_{0}\left(z\right)g_{0}\left(z\right)+v\left(b_{0}\left(z\right)g_{1}\left(z\right)+b_{1}\left(z\right)g_{0}\left(z\right)\right)+v^{2}\\ \; & (b_{0}\left(z\right)g_{2}\left(z\right)+b_{1}\left(z\right)g_{1}\left(z\right)+b_{2}\left(z\right)g_{0}\left(z\right)+c\left(z\right)a_{2}\left(z\right)). \end{array}$$Comparing both equations, we obtain $b_{0}\left(z\right)\in F_{4}\backslash \left\{0\right\}$, say $b_{0}$, this implies that $g_{0}\left(z\right)$ is self-reciprocal. Therefore, $z^{r-s}g_{1}^{*}\left(z\right)=b_{0}g_{1}\left(z\right)+b_{1}g_{0}\left(z\right)$ and $z^{r-t}g_{2}^{*}\left(z\right)=b_{0}g_{2}\left(z\right)+b_{1}g_{1}\left(z\right)+b_{2}\left(z\right)g_{0}\left(z\right)+c\left(z\right)a_{2}\left(z\right)$; this implies that $a_{2}\left(z\right)$ divides $z^{r-t}g_{2}^{*}\left(z\right)+b_{0}g_{2}\left(z\right)+b_{1}g_{1}\left(z\right)$.Again, $v^{2}a_{2}^{*}\left(z\right)\in C$ and hence $a_{2}\left(z\right)|g_{0}\left(z\right)$ implies that $a_{2}\left(z\right)$ is self-reversible.Conversely, suppose conditions (1) and (2) hold. Then
$$\begin{array}{cc} {\left(g_{0}\left(z\right)+vg_{1}\left(z\right)+v^{2}g_{2}\left(z\right)\right)}^{*} & =g_{0}^{*}\left(z\right)+vz^{r-s}g_{1}^{*}\left(z\right)+v^{2}z^{r-t}g_{2}^{*}\left(z\right)\\ \; & =b_{0}g_{0}\left(z\right)+v\left(b_{0}g_{1}\left(z\right)+b_{1}g_{0}\left(z\right)\right)+v^{2}(b_{0}g_{2}\left(z\right)\\ \; & +b_{1}g_{1}\left(z\right)+c\left(z\right)a_{2}\left(z\right))\;for\;some\;c\left(z\right)\in F_{4}\left[z\right]\\ \; & =b_{0}\left(g_{0}\left(z\right)+vg_{1}\left(z\right)+v^{2}g_{2}\left(z\right)\right)+vb_{1}(g_{0}\left(z\right)\\ \; & +vg_{1}\left(z\right)+v^{2}g_{2}\left(z\right))+c\left(z\right)v^{2}a_{2}\left(z\right)\in C. \end{array}$$Therefore, C is reversible. □

The following theorem states the reversible condition of odd length codes or a code satisfying Corollary 2.

**Theorem** **5.**
*Let $C=\langle g_{0}\left(z\right),va_{1}\left(z\right),v^{2}a_{2}\left(z\right)\rangle$ be a cyclic code over R with $a_{2}\left(z\right)|a_{1}\left(z\right)|g_{0}\left(z\right)|(z^{n}$−1). Then code C is reversible if and only if polynomials $g_{0}\left(z\right)$, $a_{1}\left(z\right)$ and $a_{2}\left(z\right)$ are self-reversible.*


**Proof.** Let C be a reversible code. Then for some polynomials $b_{0}\left(z\right),\;b_{1}\left(z\right)\;\mathrm{and}\;b_{2}\left(z\right)$ in $F_{4}\left[z\right]$, we have ${\left(g_{0}\left(z\right)\right)}^{*}=b_{0}\left(z\right)g_{0}\left(z\right)+vb_{1}\left(z\right)a_{1}\left(z\right)+v^{2}b_{2}\left(z\right)a_{2}\left(z\right).$Comparing both sides, we obtain $b_{0}\left(z\right)\in F_{4}\backslash \left\{0\right\}$, say $b_{0}$, since $degf^{*}\left(z\right)\leq degf\left(z\right)$, then $g_{0}\left(z\right)$ is self-reciprocal. Similarly, $a_{1}\left(z\right)$ and $a_{2}\left(z\right)$ are self-reciprocal polynomials.Conversely, let the polynomials $g_{0}\left(z\right)$, $a_{1}\left(z\right)$, and $a_{2}\left(z\right)$ be self-reciprocal. Then, elements of C are provided by the polynomial $b_{0}\left(z\right)g_{0}\left(z\right)+vb_{1}\left(z\right)a_{1}\left(z\right)+v^{2}b_{2}\left(z\right)a_{2}\left(z\right),$ therefore by Lemma 4, we have
$$\begin{array}{cc} {\left(b_{0}\left(z\right)g_{0}\left(z\right)+vb_{1}\left(z\right)a_{1}\left(z\right)+v^{2}b_{2}\left(z\right)a_{2}\left(z\right)\right)}^{*} & ={\left(b_{0}\left(z\right)g_{0}\left(z\right)\right)}^{*}+v{\left(b_{1}\left(z\right)a_{1}\left(z\right)\right)}^{*}z^{r-s}\\ & +v^{2}{\left(b_{2}\left(z\right)a_{2}\left(z\right)\right)}^{*}z^{r-t}.\\ \; & =b_{0}^{*}\left(z\right)g_{0}^{*}\left(z\right)+vz^{r-s}b_{1}^{*}\left(z\right)a_{1}^{*}\left(z\right)\\ & +v^{2}z^{r-t}b_{2}^{*}\left(z\right)a_{2}^{*}\left(z\right)\in C. \end{array}$$Thus, C is reversible. □

Now, in the following result, we determine the rank of a code C. The proof is followed by similar arguments as in Theorem 3 of [6].

**Theorem** **6.**
*Let C be a cyclic code of length n over R such that*

$$C=\langle g_{0}\left(z\right)+vg_{1}\left(z\right)+v^{2}g_{2}\left(z\right),va_{1}\left(z\right)+vp\left(z\right),v^{2}a_{2}\left(z\right)\rangle ,$$

*where $g_{0}\left(z\right),\;g_{1}\left(z\right),\;g_{2}\left(z\right)$, and $a_{2}\left(z\right)$ are polynomials in $F_{4}\left[z\right]$ and $deg(g_{0}\left(z\right)+vg_{1}\left(z\right)+v^{2}g_{2}\left(z\right))$=r,$deg(a_{1}\left(z\right))=s$ and $deg(a_{2}\left(z\right))=t$. Then C is a free module and rank(C)=n−t. Moreover, the basis of C is provided by the set S, where*

$$\begin{array}{cc} S= & \{\left(g_{0}\left(z\right)+vg_{1}\left(z\right)+v^{2}g_{2}\left(z\right)\right),x\left(g_{0}\left(z\right)+vg_{1}\left(z\right)+v^{2}g_{2}\left(z\right)\right),\ldots ,z^{n-r-1}(g_{0}\left(z\right)+vg_{1}\left(z\right)\\ & +v^{2}g_{2}\left(z\right)),\left(va_{1}\left(z\right)+v^{2}p\left(z\right)\right),x\left(va_{1}\left(z\right)+v^{2}p\left(z\right)\right),\ldots ,z^{r-s-1}\left(va_{1}\left(z\right)+v^{2}p\left(z\right)\right),\\ & v^{2}a_{2}\left(z\right),v^{2}xa_{2}\left(z\right),\ldots ,v^{2}z^{s-t-1}a_{2}\left(z\right))\}. \end{array}$$



## 4. Reversible Skew Cyclic Codes over R

In this part, we focus on the structure of skew cyclic codes over R and establish a necessary and sufficient condition for these codes to be reversible. We first define the skew polynomial ring over R and provide some definitions that will be used in this section.

Let $\theta \in Aut(F_{4})$ be defined by $\theta \left(a\right)=a^{2}$. Now, consider a map σ:R⟶R defined by:$$\sigma \left(a_{0}+a_{1}v+a_{2}v^{2}\right)=\theta \left(a_{0}\right)+\theta \left(a_{1}\right)v+\theta \left(a_{2}\right)v^{2},$$
where $a_{0},a_{1},a_{2}\in F_{4}$. Since σ is an extension of θ, σ is an automorphism of R. Let us consider the set:$$R\left[z;\sigma \right]=\left\{a_{0}+a_{1}z+\ldots +a_{n}z^{n}|a_{i}\in R\;\forall \;i,n\in N\right\}.$$

Define the addition on R[z;σ] as the usual addition of polynomials and multiplication under the rule $\left(a_{i}z^{i}\right)\left(a_{j}z^{j}\right)=a_{i}\sigma^{i}\left(a_{j}\right)z^{i+j}$. Then, it is easy to show that R[z;σ] forms a ring under the above binary operations, known as a skew polynomial ring. Here, $\left(a_{i}z^{i}\right)\left(a_{j}z^{j}\right)\neq \left(a_{j}z^{j}\right)\left(a_{i}z^{i}\right)$ unless σ is identity automorphism.

**Definition** **3.**
*Let $\tau_{\sigma }:R^{n}⟶R^{n}$ be a skew cyclic shift operator defined by:*

$$\tau_{\sigma }\left(a_{0},a_{1},\ldots ,a_{n-1}\right)=\left(\sigma \left(a_{n-1}\right),\sigma \left(a_{0}\right),\cdots ,\sigma \left(a_{n-2}\right)\right),\forall \;\left(a_{0},a_{1},\ldots ,a_{n-1}\right)\in R^{n}.$$

*, a linear code C of length n over R is said to be skew cyclic code if for any codeword c∈C, their skew cyclic shift $\tau_{\sigma }\left(c\right)$ belongs to C, that is, $\tau_{\sigma }\left(C\right)=C$.*


**Definition** **4.**
*For skew polynomials, a(z)andb(z)≠0, the polynomial b(z) is said to be rightly divided by a(z) if and only if there exists a skew polynomial q(z) such that a(z)=q(z)b(z) and we denote it by ${b\left(z\right)|}_{r}a\left(z\right)$.*


Using similar arguments as in the commutative case, we provide the structure of the skew cyclic codes over R for automorphism σ.

**Theorem** **7.**
*Let C be a skew cyclic code in $\frac{R[z;\sigma ]}{\langle z^{n}-1\rangle }$. Then, $C=\langle g_{0}\left(z\right)+vg_{1}\left(z\right)+v^{2}g_{2}\left(z\right),va_{1}\left(z\right)+v^{2}p\left(z\right),v^{2}a_{2}\left(z\right)\rangle$ with $a_{2}{\left(z\right)|}_{r}a_{1}\left(z\right){{|}_{r}g_{0}\left(z\right)|}_{r}\left(z^{n}-1\right)$ in $F_{4}\left[z;\theta \right]$, $a_{1}{\left(z\right)|}_{r}g_{1}\left(z\right)\left(\frac{z^{n}-1}{g_{0}\left(z\right)}\right)$ and $a_{2}\left(z\right)$ right divides $p\left(z\right)\left(\frac{z^{n}-1}{a_{1}\left(z\right)}\right)$, and $g_{2}\left(z\right)\left(\frac{z^{n}-1}{g_{0}\left(z\right)}\right)\left(\frac{z^{n}-1}{a_{1}\left(z\right)}\right)$.*


**Proof.** Consider the ring $R^{\prime }=\frac{F_{4}\left[v\right]}{\langle v^{2}\rangle }$ and $\sigma^{{}^{\prime }}\in Aut\left(R^{\prime }\right)$. For a skew cyclic code C over R, define a map $\psi_{1}:R\to R^{\prime }$ by $\psi_{1}\left(a+bv+cv^{2}\right)=a+bv\;\mathrm{where}\;a,b,c\in F$. Then, $\psi_{1}$ is a ring homomorphism that can be extended to a homomorphism $\phi :C\to \frac{R^{\prime }\left[z;\sigma^{\prime }\right]}{\langle z^{n}-1\rangle }$ defined by
$$\phi \left(c_{0}+c_{1}z+\ldots +c_{n-1}z^{n-1}\right)=\psi_{1}\left(c_{0}\right)+\psi_{1}\left(c_{1}\right)z+\ldots +\psi_{1}\left(c_{n-1}\right)z^{n-1}.$$Then $ker\left(\phi \right)=\{v^{2}r\left(z\right):r\left(z\right)\in F_{4}\left[z;\theta \right]/\langle z^{n}-1\rangle \}$.In order to determine the generators of cyclic code in $R_{n}=R\left[z,\sigma \right]/\langle z^{n}-1\rangle$, we need to know the image of ϕ which is a skew cyclic code in $R_{n}^{\prime }=R^{\prime }\left[z,\sigma_{2}\right]/\langle z^{n}-1\rangle$.Let *D* be a cyclic code in $R_{n}^{\prime }$. Now, define a map $\psi_{2}:R^{\prime }\to F_{4}$ by $\psi_{2}\left(a+ub\right)=a^{2}$. Then $\psi_{2}$ is a ring homomorphism. We extend $\psi_{2}$ to a ring homomorphism $\varphi :D\to F_{4}\left[z;\theta \right]/\langle z^{n}-1\rangle$ defined by
$$\begin{array}{c} \varphi \left(d_{0}+d_{1}z+\ldots +d_{n-1}z^{n-1}\right)=\psi_{2}\left(d_{0}\right)+\psi_{2}\left(d_{1}\right)z+\ldots +\psi_{2}\left(d_{n-1}\right)z^{n-1}. \end{array}$$
Then,
$$\begin{array}{cc} ker(\varphi ) & =\{vr^{\prime }\left(z\right):r^{\prime }\left(z\right)\;\mathrm{is}\;a\;\mathrm{skew}\;\mathrm{polynomial}\;\mathrm{in}\;F_{4}\left[z;\theta \right]/\langle z^{n}-1\rangle \}\\ \; & =\langle va_{1}\left(z\right)\rangle \;with\;a_{1}{\left(z\right)|}_{r}\left(z^{n}-1\right). \end{array}$$
Since the set image(φ) is also an ideal and hence a skew cyclic code generated by $g_{0}\left(z\right)$, where $g_{0}\left(z\right)$ right divides $\left(z^{n}-1\right)$. Therefore, $D=\langle g_{0}\left(z\right)+vg_{1}\left(z\right),va_{1}\left(z\right)\rangle$ where $a_{1}{\left(z\right)|}_{r}g_{0}\left(z\right)$ and $a_{1}{\left(z\right)|}_{r}\left(g_{1}\left(z\right)\frac{z^{n}-1}{g_{0}\left(z\right)}\right)$.Similarly, the set image(ϕ) is an ideal over $R^{\prime }$. Therefore, skew cyclic code C over R is provided by $C=\langle g_{0}\left(z\right)+vg_{1}\left(z\right)+v^{2}g_{2}\left(z\right),va_{1}\left(z\right)+v^{2}p\left(z\right),v^{2}a_{2}\left(z\right)\rangle$ with $a_{2}\left(z\right){{|}_{r}a_{1}\left(z\right)|}_{r}$$g_{0}{\left(z\right)|}_{r}\left(z^{n}-1\right)$ and $a_{1}{\left(z\right)|}_{r}\left(g_{1}\left(z\right)\frac{z^{n}-1}{g_{0}\left(z\right)}\right)$, $a_{2}{\left(z\right)|}_{r}\left(g_{1}\left(z\right)\frac{z^{n}-1}{g_{0}\left(z\right)}\right).$ □

**Definition** **5.**
*Let $g\left(z\right)=g_{0}+g_{1}z+\ldots +g_{m}z^{m}$ be a polynomial in $F_{4}\left[z,\theta \right]$. Then, g(z) is said to be a palindromic polynomial if $g_{i}=g_{m-i}$ and θ-palindromic if $g_{i}=\theta \left(g_{m-i}\right)$ where i∈{1,2,…,m}.*


Note that if the length of the code C is odd, then the skew cyclic codes and cyclic codes are equivalent (Theorem 8 in [17]). Now, we provide two lemmas to check the reversibility of the even length skew cyclic codes over the field $F_{4}$.

**Lemma** **5.**
*Let C be a skew cyclic code of even length generated by a monic polynomial $f\left(z\right)=1+f_{1}z+\ldots +f_{m-1}z^{m-1}+z^{m}$ of even degree, where ${f\left(z\right)|}_{r}\left(z^{n}-1\right)$ in $F_{4}\left[z,\theta \right]$. Then, the code C is reversible if and only if skew polynomial f(z) is θ-palindromic.*


**Proof.** Let C be a skew cyclic code of even length generated by the θ-palindromic polynomial f(z) of even degree *m* over the ring $F_{4}$. Then, the elements of the generated code are provided by $\sum_{i=0}^{k-1}\alpha_{i}z^{i}f\left(z\right)$. From the repetitive use of Lemma 3, for $c=\phi (\sum_{i=0}^{k-1}\alpha_{i}z^{i}f\left(z\right))\in C$, we obtain:
$${\left(\phi \left(\sum_{i=0}^{k-1}\alpha_{i}z^{i}f\left(z\right)\right)\right)}^{r}=\phi \left(\sum_{i=0}^{k-1}\alpha_{i}z^{k-i-1}f\left(z\right)\right)\in C.$$
where $\alpha \in F_{4}$ and k=n−m. Since $c^{r}$ belongs to the code C, C is a reversible code.Conversely, let C be a reversible code generated by $f\left(z\right)=1+f_{1}z+\ldots +f_{m-1}z^{m-1}+z^{m}$. Then, because n−m−1 is odd:
$$z^{n-m-1}f\left(z\right)=z^{n-m-1}+\theta \left(f_{1}\right)z^{n-m}+\ldots +\theta \left(f_{m-1}\right)z^{n-2}+z^{n-1}.$$
Since C is a skew cyclic and reversible code,
$$\begin{array}{c} {\left[z^{n-m-1}f\left(z\right)\right]}^{r}=1+\theta \left(f_{m-1}\right)z+\theta \left(f_{m-2}\right)z^{2}+\ldots +\theta \left(f_{1}\right)z^{m-1}+z^{m}\in C. \end{array}$$
Further, we obtain $deg(f\left(z\right)-{\left[z^{n-m-1}f\left(z\right)\right]}^{r})<m$, which contradicts the fact that f(z) is a minimal degree polynomial in C implies $f\left(z\right)-{\left[z^{n-m-1}f\left(z\right)\right]}^{r}=0$. Comparing coefficients, we obtain:
$$[f_{i}-\theta \left(f_{m-i}\right)]=0$$
for i=1,…,m−1. Thus, $f_{i}=\theta \left(f_{m-i}\right)$ and the polynomial f(z) is θ-palindromic. □

**Lemma** **6.**
*Let C be a skew cyclic code of even length generated by a monic polynomial $f\left(z\right)=1+f_{1}z+\ldots +f_{m-1}z^{m-1}+z^{m}$ of odd degree, where ${f\left(z\right)|}_{r}\left(z^{n}-1\right)$ in $F_{4}\left[z,\theta \right]$. Then, the code C is reversible if and only if the skew polynomial f(z) is palindromic.*


**Proof.** Let C be a skew cyclic code of even length generated by a palindromic polynomial f(z) of odd degree *m* over the ring $F_{4}$. Then, elements of the generated code are provided by $\sum_{j=0}^{k-1}\alpha_{j}z^{j}f\left(z\right)$. From the repetitive use of Lemma 3 and using the property of the palindromic polynomial, for $C=\phi (\sum_{j=0}^{k-1}\alpha_{j}z^{j}f\left(z\right))\in C$, we obtain:
$${\left(\phi \left(\sum_{j=0}^{k-1}\alpha_{j}z^{j}f\left(z\right)\right)\right)}^{r}=\phi \left(\sum_{j=0}^{k-1}\alpha_{j}z^{k-j-1}f\left(z\right)\right)\in C.$$
where $\alpha \in F_{4}$ and k=n−m. Since the reverse of C belongs to C, the code C is reversible.
Conversely, let C be a reversible code generated by $f\left(z\right)=1+f_{1}z+\ldots +f_{m-1}z^{m-1}+z^{m}$. Since n−m−1 is even:
$$z^{n-m-1}f\left(z\right)=z^{n-m-1}+f_{1}z^{n-m}+\ldots +f_{m-1}z^{n-2}+z^{n-1}.$$Furthermore, the code C is a skew cyclic as well as reversible code; therefore, ${\left[z^{n-m-1}f\left(z\right)\right]}^{r}$∈C and:
$${\left[z^{n-m-1}f\left(z\right)\right]}^{r}=\left[1+f_{m-1}z+f_{m-2}z^{2}+\ldots +f_{1}z^{m-1}+z^{m}\right]\in C.$$
This implies that $deg(f\left(z\right)-{\left[z^{n-m-1}f\left(z\right)\right]}^{r})<m$, which contradicts the fact that f(z) is a minimal degree polynomial in C. Hence, $f\left(z\right)-{\left[z^{n-m-1}f\left(z\right)\right]}^{r}=0$. By comparing the coefficients, we obtain
$$\left[f_{i}-f_{m-i}\right]=0\;\mathrm{and}\;f_{i}=f_{m-i},$$
for i=1,…,m−1. Thus, the given polynomial f(z) is palindromic. □

Now, in the next theorem, we provide necessary and sufficient conditions for a skew cyclic code C to be reversible in terms of palindromic and θ-palindromic polynomials. These conditions depend on the degree of generator polynomials of C.

**Theorem** **8.**
*Let $C=\langle g_{0}\left(z\right),vg_{1}\left(z\right),v^{2}g_{2}\left(z\right)\rangle$ be a skew cyclic code of even length, where $g_{i}\left(z\right)$ right divides $\left(z^{n}-1\right)$ in $F_{4}\left[z,\theta \right]$ and $deg(g_{i}\left(z\right))$ is even (odd), for i=0,1,2. Then, the code C is reversible if and only if skew polynomials $g_{i}\left(z\right)$ are θ-palindromic (palindromic) for i=0,1,2.*


## 5. DNA Codes over R

In this section, we discuss the complementary condition of the codes obtained from previous sections to obtain DNA codes. For a DNA code, the reversible and complement conditions are essential [21].

**Definition** **6.**
*Let C be a code of length n over R. If $\Phi {\left(C\right)}^{rc}\in \Phi \left(C\right)$ for all c∈C, then C or equivalently Φ(C) is called a DNA code.*


In the following lemma, we provide some relations on ring elements and their complement using the Gray map provided in Equation (Equation 1).

**Lemma** **7.**
*For the given cyclic code in Section 3, the following conditions hold:*
*(1)* 
*For any $r\in R,\;r+r^{c}=v^{2}.$*
*(2)* 
*For any $r_{1},r_{2}\in R,\;r_{1}^{c}+r_{2}^{c}={\left(r_{1}+r_{2}\right)}^{c}+v^{2}$.*



**Proof.** This lemma can easily be proved by observing Table 1. □

**Remark** **2.**
*We identify i(z) by the polynomial $1+z+z^{2}+\cdots +z^{n-1}.$*


**Theorem** **9.**
*Given a polynomial a(z) in R[z]. We have $a{\left(z\right)}^{rc}=a{\left(z\right)}^{r}+v^{2}i\left(z\right)$.*


**Proof.** Let C be a reversible-complement code. Then, by definition, C is reversible and 0∈C implies that ${\left(0+0z+\ldots +0z^{n-1}\right)}^{c}\in C$. That is, C is reversible and $v^{2}+v^{2}z+\ldots +v^{2}z^{n-1}\in C$.Conversely, let $a\left(z\right)=a_{0}+a_{1}z+\ldots +a_{n-1}z^{n-1}+a_{n}z^{n}$ be a polynomial in R[z]. Then:
$$\begin{array}{cc} a{\left(z\right)}^{rc} & =a_{n}^{c}+a_{n-1}^{c}z+\ldots +a_{1}^{c}z^{n-1}+a_{0}^{c}z^{n}\\ \; & =a_{n}+v^{2}+\left(a_{n-1}+v^{2}\right)z+\left(a_{n-2}+v^{2}\right)z^{2}+\ldots \\ \; & +\left(a_{1}+v^{2}\right)z^{n-1}+\left(a_{0}+v^{2}\right)z^{n}\\ \; & =v^{2}i\left(z\right)+a{\left(z\right)}^{r}\in C. \end{array}$$
Thus, cyclic code C is a reversible-complement code. □

**Corollary** **4.**
*Let C be a cyclic code of even length over R. Then, C is a DNA code if and only if C is reversible and $v^{2}i\left(z\right)$ is in C.*


**Proof.** It is obvious from above theorem. □

## 6. Computational Results

Now, we provide some examples of DNA codes satisfying the above-mentioned constraints. We consider DNA code of any length (even or odd). All the computational works are performed by using Magma software [20].

**Example** **1.**
*In $F_{4}\left[z\right]$, we have:*

$$z^{6}-1={\left(z+1\right)}^{2}{\left(z+t\right)}^{2}{\left(z+t^{2}\right)}^{2}.$$

*Let C be a cyclic code of length n=6 over R provided by:*

$$C=\langle z^{4}+z^{2}+1,v\left(z^{4}+z^{2}+1\right),v^{2}\left(z^{4}+z^{2}+1\right)\rangle .$$

*Then, using Theorem 2, we obtain d(C)=3. Furthermore, (x−1) does not divide $\left(z^{4}+z^{2}+1\right)$ and polynomial $\left(z^{4}+z^{2}+1\right)$ is self reciprocal. Thus, we obtain a DNA code C of parameters $\left(18,4^{6},3\right)$.*


In the next example, we provide some DNA codes of arbitrary lengths that are generated from cyclic codes over R.

**Example** **2.**
*Suppose C is a cyclic code of the form $C=\langle g_{0}\left(z\right)+vg_{1}\left(z\right)+v^{2}g_{2}\left(z\right),va_{1}\left(z\right)+v^{2}p\left(z\right),v^{2}a_{2}\left(z\right)\rangle ,$ where $\gcd(\frac{z^{n}-1}{a_{2}\left(z\right)},g_{0}\left(z\right))=1$. If $g_{0}\left(z\right)=a_{1}\left(z\right)=a_{2}\left(z\right)$, then we list several DNA codes in Table 2 that are obtained from cyclic code C. Since $g_{0}\left(z\right)$, $a_{1}\left(z\right)$, and $a_{2}\left(z\right)$ are equal, therefore, in Table 2, we mention only $g_{0}\left(z\right)$. For brevity, polynomial $z^{2}+b_{1}z+b_{0}$ is represented as $b_{0}b_{1}1$.*


**Example** **3.**
*Consider a cyclic code C of length n=9 over ring R. In $F_{4}\left[z\right]$, we have:*

$$z^{9}-1=\left(z+1\right)\left(z+t\right)\left(z+t^{2}\right)\left(z^{3}+t\right)\left(z^{3}+t^{2}\right).$$

*To write briefly, we identify factors by $g_{1},g_{2},g_{3},g_{4}$, and $g_{5}$, respectively. The codes for n=9 are provided in Table 3. All the codes are better than the codes that appeared in [14].*


**Example** **4.**
*Consider a cyclic code C of length n=15 over ring R. In $F_{4}\left[z\right]$, we have*

$$\begin{array}{cc} z^{15}-1= & \left(z+1\right)\left(z+t\right)\left(z+t^{2}\right)\left(z^{2}+z+t\right)\left(z^{2}+z+t^{2}\right)\left(z^{2}+tz+1\right)\\ \; & \left(z^{2}+tz+t\right)\left(z^{2}+t^{2}z+1\right)\left(z^{2}+t^{2}z+t^{2}\right). \end{array}$$

*For brevity, we identify the factors by $g_{1},g_{2},g_{3},g_{5},g_{6},g_{7},g_{8},\;\mathrm{and}\;g_{9}$, respectively. DNA codes for n=15 are provided in Table 4. All the obtained DNA codes are better than the codes provided in [14].*

*In particular, if $C=\langle g_{2}g_{3}g_{4}g_{5}g_{6}g_{7}g_{8}g_{9},\;vg_{2}g_{3}g_{4}g_{5}g_{6}g_{7}g_{8}g_{9},\;v^{2}g_{2}g_{3}g_{4}g_{5}g_{6}g_{7}g_{8}g_{9}\rangle$, then we obtain a DNA code with parameters $\left[45,4^{3},15\right]$. Further, we list all the DNA codewords of the obtained DNA code in Table 5. Furthermore, the edit distance of the obtained DNA code is 2, given by the codewords “TCCTCCTCCTCCTCCTCCTCCTCCTCC" and “CTCCTCCTCCTCCTCCTCCTCCTCCTC".*


## 7. Conclusions

In this paper, we have studied reversible and DNA codes using the chain ring $R=F_{4}\left[v\right]/\langle v^{3}\rangle$. We have defined a Gray map on R and found codons corresponding to the elements of R. In this way, we have obtained good DNA and reversible codes with the Hamming distances. In the future, one can work on DNA codes over a generalized structure of R as well as DNA codes by using skew polynomial rings.

## Figures and Tables

**Table 1 entropy-25-00239-t001:** Codons correspondence with the elements of R.

0	AAA	$v^{2}$	TTT	$tv^{2}$	GGG	$t^{2}v^{2}$	CCC
1	TAA	$v^{2}+1$	ATT	$tv^{2}+1$	CGG	$t^{2}v^{2}+1$	GCC
*t*	GAA	$v^{2}+t$	CTT	$tv^{2}+t$	AGG	$t^{2}v^{2}+t$	TCC
$t^{2}$	CAA	$v^{2}+t^{2}$	GTT	$tv^{2}+t^{2}$	TGG	$t^{2}v^{2}+t^{2}$	ACC
*v*	TTA	$v^{2}+v$	AAT	$tv^{2}+v$	CCG	$t^{2}v^{2}+v$	GGC
v+1	ATA	$v^{2}+v+1$	TAT	$tv^{2}+v+1$	GCG	$t^{2}v^{2}+v+t$	AGC
v+t	CTA	$v^{2}+v+t$	GAT	$tv^{2}+v+t$	TCG	$t^{2}v^{2}+v+1$	CGC
$v+t^{2}$	GTA	$v^{2}+v+t^{2}$	CAT	$tv^{2}+v+t^{2}$	ACG	$t^{2}v^{2}+v+t^{2}$	TGC
tv	GGA	$v^{2}+tv$	CCT	$tv^{2}+tv$	AAG	$t^{2}v^{2}+tv$	TTC
tv+1	CGA	$v^{2}+tv+1$	GCT	$tv^{2}+tv+1$	TAG	$t^{2}v^{2}+tv+1$	ATC
tv+t	AGA	$v^{2}+tv+t$	TCT	$tv^{2}+tv+t$	GAG	$t^{2}v^{2}+tv+t$	CTC
$tv+t^{2}$	TGA	$v^{2}+tv+t^{2}$	ACT	$tv^{2}+tv+t^{2}$	CAG	$t^{2}v^{2}+tv+t^{2}$	GTC
$t^{2}v$	CCA	$v^{2}+t^{2}v$	GGT	$tv^{2}+t^{2}v$	TTG	$t^{2}v^{2}+t^{2}v$	AAC
$t^{2}v+1$	GCA	$v^{2}+t^{2}v+1$	CGT	$tv^{2}+t^{2}v+1$	ATG	$t^{2}v^{2}+t^{2}v+1$	TAC
$t^{2}v+t$	TCA	$v^{2}+t^{2}v+t$	AGT	$tv^{2}+t^{2}v+t$	CTG	$t^{2}v^{2}+t^{2}v+t$	GAC
$t^{2}v+t^{2}$	ACA	$v^{2}+t^{2}v+t^{2}$	TGT	$tv^{2}+t^{2}v+t^{2}$	GTG	$t^{2}v^{2}+t^{2}v+t^{2}$	CAC

**Table 2 entropy-25-00239-t002:** DNA codes of different lengths.

Length	$g_{0}\left(z\right)$	Type of Code	Gray Image
5	1t1	(5,3,3)	$\left(15,4^{9},3\right)$
5	11111	(5,1,5)	$\left(15,4^{3},5\right)$
6	10101	(6,2,3)	$\left(18,4^{6},3\right)$
10	101010101	(10,2,5)	$\left(30,4^{6},5\right)$
13	1t0(1+t)0t1	(13,7,5)	$\left(39,4^{21},5\right)$
14	1010101010101	(14,2,7)	$\left(42,4^{6},7\right)$
17	11t11	(17,13,4)	$\left(51,4^{39},4\right)$
17	1(1+t)11t11(1+t)1	(17,9,7)	$\left(51,4^{27},7\right)$
29	1t0t(1+t)1(1+t)t(1+t)1(1+t)t0t1	(10,1,5)	$\left(30,4^{3},5\right)$

**Table 3 entropy-25-00239-t003:** Codes of length 27.

Sr No	Generator of Code	Type of Code	Gray Image	DNA Code [14]
1	$\langle g_{2}g_{3},vg_{2}g_{3},v^{2}g_{2}g_{3}\rangle$	(9,7,2)	$\left(27,4^{21},2\right)$	$\left(27,4^{14},2\right)$
2	$\langle g_{4}g_{5},vg_{4}g_{5},v^{2}g_{4}g_{5}\rangle$	(9,3,3)	$\left(27,4^{9},3\right)$	$\left(27,4^{6},3\right)$
3	$\langle g_{2}g_{3}g_{4}g_{5},vg_{2}g_{3}g_{4}g_{5},v^{2}g_{2}g_{3}g_{4}g_{5}\rangle$	(9,1,9)	$\left(27,4^{3},9\right)$	$\left(27,4^{2},9\right)$

**Table 4 entropy-25-00239-t004:** Codes of length 45.

Code	Type of Code	Gray Image	DNA Code [14]
$\langle g_{2}g_{3},vg_{2}g_{3},v^{2}g_{2}g_{3}\rangle$	(15,13,2)	$\left(45,4^{39},2\right)$	$\left(45,4^{26},2\right)$
$\langle g_{2}g_{3}g_{6},vg_{2}g_{3}g_{6},v^{2}g_{2}g_{3}g_{6}\rangle$	(15,11,4)	$\left(45,4^{33},4\right)$	$\left(45,4^{24},3\right)$
$\langle g_{4}g_{8}g_{9},vg_{4}g_{8}g_{9},v^{2}g_{4}g_{8}g_{9}\rangle$	(15,9,5)	$\left(45,4^{27},5\right)$	$\left(45,4^{18},5\right)$
$\langle g_{2}g_{3}g_{5}g_{6}g_{7},vg_{2}g_{3}g_{5}g_{6}g_{7},v^{2}g_{2}g_{3}g_{5}g_{6}g_{7}\rangle$	(15,7,7)	$\left(45,4^{21},7\right)$	$\left(45,4^{14},7\right)$
$\langle g_{2}g_{3}g_{4}g_{5}g_{6}g_{7}g_{9},vg_{2}g_{3}g_{4}g_{5}g_{6}g_{7}g_{9},v^{2}g_{2}g_{3}g_{4}g_{5}g_{6}g_{7}g_{9}\rangle$	(15,3,9)	$\left(45,4^{9},9\right)$	$\left(45,4^{6},9\right)$

**Table 5 entropy-25-00239-t005:** Codewords of length 45 and dimension 3.

AAAAAAAAAAAAAAAAAAAAAAAAAAA	TAATAATAATAATAATAATAATAATAA
GAAGAAGAAGAAGAAGAAGAAGAAGAA	CAACAACAACAACAACAACAACAACAA
TTATTATTATTATTATTATTATTATTA	ATAATAATAATAATAATAATAATAATA
CTACTACTACTACTACTACTACTACTA	GTAGTAGTAGTAGTAGTAGTAGTAGTA
GGAGGAGGAGGAGGAGGAGGAGGAGGA	CGACGACGACGACGACGACGACGACGA
AGAAGAAGAAGAAGAAGAAGAAGAAGA	TGATGATGATGATGATGATGATGATGA
CCACCACCACCACCACCACCACCACCA	GCAGCAGCAGCAGCAGCAGCAGCAGCA
TCATCATCATCATCATCATCATCATCA	ACAACAACAACAACAACAACAACAACA
TTTTTTTTTTTTTTTTTTTTTTTTTTT	ATTATTATTATTATTATTATTATTATT
CTTCTTCTTCTTCTTCTTCTTCTTCTT	GTTGTTGTTGTTGTTGTTGTTGTTGTT
AATAATAATAATAATAATAATAATAAT	TATTATTATTATTATTATTATTATTAT
GATGATGATGATGATGATGATGATGAT	CATCATCATCATCATCATCATCATCAT
CCTCCTCCTCCTCCTCCTCCTCCTCCT	GCTGCTGCTGCTGCTGCTGCTGCTGCT
TCTTCTTCTTCTTCTTCTTCTTCTTCT	ACTACTACTACTACTACTACTACTACT
GGTGGTGGTGGTGGTGGTGGTGGTGGT	CGTCGTCGTCGTCGTCGTCGTCGTCGT
AGTAGTAGTAGTAGTAGTAGTAGTAGT	TGTTGTTGTTGTTGTTGTTGTTGTTGT
GGGGGGGGGGGGGGGGGGGGGGGGGGG	CGGCGGCGGCGGCGGCGGCGGCGGCGG
AGGAGGAGGAGGAGGAGGAGGAGGAGG	TGGTGGTGGTGGTGGTGGTGGTGGTGG
CCGCCGCCGCCGCCGCCGCCGCCGCCG	GCGGCGGCGGCGGCGGCGGCGGCGGCG
TCGTCGTCGTCGTCGTCGTCGTCGTCG	ACGACGACGACGACGACGACGACGACG
AAGAAGAAGAAGAAGAAGAAGAAGAAG	TAGTAGTAGTAGTAGTAGTAGTAGTAG
GAGGAGGAGGAGGAGGAGGAGGAGGAG	CAGCAGCAGCAGCAGCAGCAGCAGCAG
TTGTTGTTGTTGTTGTTGTTGTTGTTG	ATGATGATGATGATGATGATGATGATG
CTGCTGCTGCTGCTGCTGCTGCTGCTG	GTGGTGGTGGTGGTGGTGGTGGTGGTG
CCCCCCCCCCCCCCCCCCCCCCCCCCC	GCCGCCGCCGCCGCCGCCGCCGCCGCC
TCCTCCTCCTCCTCCTCCTCCTCCTCC	ACCACCACCACCACCACCACCACCACC
GGCGGCGGCGGCGGCGGCGGCGGCGGC	AGCAGCAGCAGCAGCAGCAGCAGCAGC
CGCCGCCGCCGCCGCCGCCGCCGCCGC	TGCTGCTGCTGCTGCTGCTGCTGCTGC
TTCTTCTTCTTCTTCTTCTTCTTCTTC	ATCATCATCATCATCATCATCATCATC
CTCCTCCTCCTCCTCCTCCTCCTCCTC	GTCGTCGTCGTCGTCGTCGTCGTCGTC
AACAACAACAACAACAACAACAACAAC	CACCACCACCACCACCACCACCACCAC
TACTACTACTACTACTACTACTACTAC	GACGACGACGACGACGACGACGACGAC

## Data Availability

The data that support the findings of this study are available from the corresponding author upon reasonable request.

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
