# Peer review of "DNA Code from Cyclic and Skew Cyclic Codes over F4[v]/v3"

_entropy, 2023, doi:10.3390/e25020239_

Round 1

Reviewer 1 Report

See the attachment

Author Response

Thank you for the constructive comments, we have revised our paper accordingly. Please find the attached file for our detailed responses.

Reviewer 2 Report

The paper studied some structural properties of cyclic and skew cyclic codes over the finite chain ring F4[v]/(v3). They gave a Gray map. Under the Gray map, they obtained some new DNA codes. In general, this paper is well written. In my opinion, it can be accepted by the journal Entropy after some minor corrections. Most of them are typos. 

1. Page 1, Line 21: Linear Codes ---> Linear codes; 2. Page 2, Line 58: (a_0, a_1...a_{n-1}) ---> (a_0, a_1, ..., a_{n-1}). Please check the whole paper. There are so many same typos. 3. Page 2, Line 68: gray ---> Gray; 4. Page 3, Line 91: Codes --> codes; 5.  In this paper, the authors claimed that they have obtained some new DNA codes. However, I can not see any comparation of them. In other words, the authors should explian why the DNA codes they obtained are new.

Author Response

(The authors gave the same response as above.)
